# Imaging Flow Cytometry to Study Biofilm-Associated Microbial Aggregates

**DOI:** 10.3390/molecules26237096

**Published:** 2021-11-24

**Authors:** Michał Konieczny, Peter Rhein, Katarzyna Czaczyk, Wojciech Białas, Wojciech Juzwa

**Affiliations:** 1Department of Biotechnology and Food Microbiology, Poznan University of Life Sciences, ul. Wojska Polskiego 28, 60-627 Poznan, Poland; mkonieczny@luminexcorp.com (M.K.); katarzyna.czaczyk@up.poznan.pl (K.C.); wojciech.bialas@up.poznan.pl (W.B.); 2Luminex B.V., A DiaSorin Company, 5215 MV ‘s-Hertogenbosch, The Netherlands; prhein@luminexcorp.com

**Keywords:** food-processing, biofilm dispersal, single-cell analysis, bioimaging, machine learning

## Abstract

The aim of the research was to design an advanced analytical tool for the precise characterization of microbial aggregates from biofilms formed on food-processing surfaces. The approach combined imaging flow cytometry with a machine learning-based interpretation protocol. Biofilm samples were collected from three diagnostic points of the food-processing lines at two independent time points. The samples were investigated for the complexity of microbial aggregates and cellular metabolic activity. Thus, aggregates and singlets of biofilm-associated microbes were simultaneously examined for the percentages of active, mid-active, and nonactive (dead) cells to evaluate the physiology of the microbial cells forming the biofilm structures. The tested diagnostic points demonstrated significant differences in the complexity of microbial aggregates. The significant percentages of the bacterial aggregates were associated with the dominance of active microbial cells, e.g., 75.3% revealed for a mushroom crate. This confirmed the protective role of cellular aggregates for the survival of active microbial cells. Moreover, the approach enabled discriminating small and large aggregates of microbial cells. The developed tool provided more detailed characteristics of bacterial aggregates within a biofilm structure combined with high-throughput screening potential. The designed methodology showed the prospect of facilitating the detection of invasive biofilm forms in the food industry environment.

## 1. Introduction

Instrumental methods, assessing the complexity of microbial populations, with particular emphasis on differences in the physiological state of individual cells, constitute a significant breakthrough in microbiological analyses [1]. These methods enrich the identification procedures used so far, with the assessment of the activity of all identified species or even individual representatives of these species. This is of particular importance among others for the effective detection of the microbiological contamination of products or for the evaluation of the suitability of starter cultures before their use in the production process [2].

Complex bacterial communities exist within biofilms, including those formed on industrial food-processing lines. Food industry environments are prone to biofilm formation [3]. This is due to nutrient-rich food matrix components covering food contact surfaces combined with long production periods and large working surfaces [4,5,6]. Food-processing environments can be colonized by a wide variety of bacterial species, which can contribute to the formation of biofilm structures. This process is based on interactions within microbial communities to constitute a complex and dynamic network shaping biofilm architecture responsible for specific functions [7,8,9,10]. Multiple studies have demonstrated that multispecies biofilms are less sensitive to antimicrobial agents than their monospecies counterparts [11,12,13,14,15,16,17,18].

Biofilm-associated microbial cells demonstrate considerable structural and functional diversities, manifested by the existence of a number of microenvironments within the entire structure of the biofilm. These ecological niches differ in terms of pH, oxygen concentration, nutrient availability, and microbial cell density [19]. Thus, the biofilm is characterized by a heterogenic distribution of diver types and species of microorganisms in a three-dimensional structure. This diversity is also evident when studying cellular parameters, especially functional ones such as the membrane potential and metabolic activity. Functional differentiation in the structure of the biofilm is mainly expressed by the presence of numerous bacterial cells, characterized by reduced metabolic activity, described as dormant or damaged as a result of contact with biocidal substances [20,21]. Furthermore, microbial cells thriving in internal biofilm fragments and characterized by low or intermediate levels of metabolic activity seem particularly challenging. Due to limited exposure (protective effect from extracellular substances and the surrounding cells), those cells may be unavailable to antimicrobial substances [22]. The occurrence of this type of cell during the biofilm life cycle is limited not only to the basic biofilm structure but mainly refers to the key invasive forms related to the late stages of biofilm maturation—microbial aggregates [23]. 

A biofilm dispersing into bacterial aggregates is initiated by the microorganisms themselves, mainly as a result of the activation of the quorum sensing mechanism. The quorum sensing mechanism is a coordinated regulation of gene expression. It controls the entire microbial population through cellular density-dependent intercellular communication [24]. As a result, a complex ecosystem such as a three-dimensional biofilm architecture may induce a dispersal phenotype of bacterial cells [25,26,27]. Thus, the non-surface-attached aggregates of microbial cells may function as “rescue capsules”, which independently or in a response to stressful conditions allow bacteria to spread effectively in the environment [28]. Moreover, the process of biofilm dispersal is one of the stages of their development, and like others, it is regulated by environmental signals, signal transduction pathways, and the so-called cellular and extracellular effectors [29]. In industrial conditions, the processes are initiated by: (i) biofilm-forming microorganisms themselves, (ii) physical factors such as shear forces, and (iii) the flow of liquid and (iv) mechanical interventions—cleaning/disinfection procedures [25].

The presence of biofilms on various technological surfaces that come into contact with food is a constant source of microbial contamination [30]. In the context of increasing the role of industrial processes in world food production, the problem of biofilm dispersal in industrial conditions requires separate attention [3]. In addition, a number of scientific works describe medical issues and the clinical aspects of the spread of biofilms as a pathogen transmission pathway in the human body. Hence, an attempt to develop an analytical tool for monitoring invasive biofilm forms—bacterial aggregates—is a challenge from the scientific and application points of view.

The aim of the research was to design an advanced tool as a combination of the analytical protocol (the use of imaging flow cytometry) and interpretation protocol (the use of unique features of the software for an advanced analysis of the cytometric results). The application of the developed tool will enable the detailed characterization of aggregates of microbial cells and the rapid screening of samples from industrial food-processing surfaces. The main impact of the designed methodology is the prospect of facilitating the detection of invasive biofilm forms in the food industry environment.

## 2. Results

The analysis of samples, constituting fragments of biofilms formed on the surfaces of food-processing technological lines, has focused on developing a protocol for interpreting the results obtained with the use of an Amnis^®^ FlowSight^®^ imaging flow cytometer. The interpretation protocol was based on the machine learning (ML) module, a feature of Amnis^®^ FlowSight^®^ instrument IDEAS^®^ software. ML automatically creates a classifier that differentiates the population of objects (e.g., single cells and aggregates) from the user input (batch data). The ML module is designed to simplify the analysis by allowing users to visually build a population to improve their discrimination and to combine multiple fluorochromes and multiple morphological parameters into one “super parameter” (super feature). The mechanism of proceeding with the use of the machine learning module is as follows: (i) the selection of the classifiers, (ii) an indication of the representative populations (through the manual selection of the images of objects), (iii) the selection of parameters with the highest separating force using machine learning using the linear discriminant analysis (LDA), and (iv) creating a “super parameter” to discriminate between the indicated populations.

As a result of the use of advanced tools for processing and interpreting the results of cytometric analyses obtained on an instrument with bioimaging, a tool was developed for the precise characterization of the degree of agglomeration of cells forming biofilms. Based on the “truth” populations specified by the user, the ML module generated the so-called super parameter (super feature classifier) composed of several dozen separate parameters that, to varying degrees (weight), determine the morphological differences within the population of the analyzed objects/cells. The classifier (super parameter) designed in the ML module allowed to precisely distinguish single microbial cells (singlets) from their complexes (aggregates) in the form of small aggregates and large aggregates composed of two to three or more than three microbial cells, respectively (Figure 1). Thereby, the classifiers reflect the complexity of the identified objects, which is difficult to achieve by a manual selection and combination of single features. A simpler S_classifier constitutes 10 highly weighted features, mainly describing the size and roundness of singlets. In contrast, for a complex structure of small and large aggregates, the ML module combines tens to hundreds of features by minimizing the prediction error.

The analysis of environmental samples at various diagnostic points and two independent experiments (time points) showed different physiological patterns among the tested biofilms derived from food-processing surfaces. An imaging flow cytometric analysis allowed detecting differences in the degree of cellular aggregation of biofilm-forming microorganisms derived from industrial surfaces and correlating them with the assessment of the metabolic activity measured as the cellular redox potential (RSG). A greater proportion of bacterial aggregates correlated with the higher metabolic activity of microorganisms forming the biofilm structure. Figure 2 provides evidence for the protective role of cellular aggregates, as the microbial cells living inside the aggregates demonstrated the high levels of metabolic activity. This effect is related to the degree of aggregation and is enhanced by the presence of large aggregates of microbial cells (Figure 2). As demonstrated in Figure 3, the diagnostic points showing significant percentages of bacterial aggregates (large and small) in a mushroom crate revealed simultaneously the dominance of active forms in both singlets and aggregates — 83 and 75.3%, respectively (mean values for both time points).

The in-depth evaluation of the complexity of microbial aggregates enabled discriminating between large and small bacterial aggregated forms. This was demonstrated by samples rich in microbial aggregates (e.g., a mushroom crate) where the percentage of large aggregates reached 17.5% in the second time point—20 September 2019, with the total percentage of agglomerates reaching 37.4% of all the observed objects. Other two diagnostic points (conveyor belt and mushroom feeder) showed lower percentages of large aggregates vs. all the aggregates: 11.3 vs. 20.6% and 16.6 vs. 25.6% for the conveyor belt and mushroom feeder, respectively (Figure 3).

In addition, individual samples showed, apart from quantitative differences (percentage of large aggregates and all aggregates), a significant qualitative differentiation in the form of the aggregate complexity (aggregates consisting of dozens of microbial cells visible in Figure 2—the mushroom crate diagnostic point). Again, the sizes of the microbial cell agglomerates correlated with an increased proportion of metabolically active cell subpopulations, which confirms the protective role of microbial cell aggregates. Figure 3 presents a detailed comparison of the analyzed samples (three diagnostic points analyzed at two time points) in terms of: (i) the degree of agglomeration (detection of single cells (singlets), small aggregates, and large aggregates of microbial cells and (ii) the metabolic activity of single cells, as well as small and large aggregates (separation of active, intermediate, and inactive (dead) forms of microorganisms). Figure 3 revealed significant variations in the results of the agglomeration degree in the tested biofilm samples; the tested diagnostic points showed different patterns of complexity of the detected agglomerates. 

Our approach demonstrated the potential of IFC in an in-depth study of the structural and functional complexities within biofilm-derived microbial aggregates. IFC, in combination with a machine learning (ML) software tool, enabled a multiparametric and morphological analysis of thousands of individual microbial cells and cellular aggregates (objects) to correlate conventional flow cytometric data with microscopic images of each individual object. IFC provided an alternative gating strategy using the aspect ratio and area features derived from cellular images [31]. Combined with ML, it increased the accuracy of the morphological study of the complexities of cellular aggregates and facilitated the compilation of these data with the physiology of individual microbial cells. Real alternatives for studying biofilms, including microscopy-based techniques like confocal laser scanning microscopy (CLSM), require the creation of a large library of images to provide statistically relevant data. Thus, they are time-consuming [32]. The developed approach emerged as an alternative for the use of powerful and user-friendly software tools for biofilm image analyses like COMSTAT [33] or BiofilmQ [34]. 

As demonstrated in our study, a single instrument may be applied to gain insight into the complexity of biofilms’ invasive forms—aggregates of bacterial cells. The aggregated forms of microbial cells forming biofilm structures were assessed in terms of the quantitative (percentages of singlets vs. small and large aggregates) and qualitative (complexity of large aggregates) diversities. A unique feature of the designed tool is the ability to distinguish within aggregates (regardless of the degree of their complexity) cells characterized by intermediate metabolic activity from those objects that consist of both active and inactive cells, which, by conventional instruments, are classified as an intermediate subpopulation. The implementation of the IFC tool enabled collecting statistically significant data and processing them using the ML-based interpretation protocol in an incomparably shorter time. The designed tool enables the fast and reliable screening of samples from working surfaces in the food industry to detect biofilm = related bacterial aggregates. It also provides key information concerning the stages of biofilm development.

## 3. Discussion

The detection and functional analysis of bacteria in biofilms by flow cytometry is often hampered by their small size and the ability to form small and large aggregates [35]. Traditional flow cytometers that use photomultiplier tubes (PMT) to collect scatter and fluorescence intensities are limited to identifying single cells and doublets based on the signal pulse geometry. Single cells have a defined relationship between the pulse area (related to the cell size), height (intensity), and width (related to the time the cell spends in the laser beam), where an increase in pulse intensity is proportional to the pulse area, which is in contrast to doublets and aggregates, which deviate from this relationship [36]. However, this process is error-prone when aggregates mimic single cells in terms of signal widths/durations due to their orientation in line or coplanar to the laser beam [37], which is even more evident for bacteria that can form clusters of different sizes and shapes [23]. Moreover, the smallest bacteria are in the submicron range similar to the size of the debris and often close to the instrumentation noise of PMT-based flow cytometers [35]. The Amnis® brand imaging flow cytometers (IFC) help to overcome these obstacles by combining the statistical power of flow cytometry with the imaging content of microscopy in one system. Unlike traditional flow cytometers, an IFC uses a charge-coupled device camera (CCD) to collect multiple high-resolution images of every cell in a flow, including the brightfield (BF), darkfield (SSC), and up to 10 fluorescent markers.

The concept of an image analysis is based on using “masks” and “features”. Masks define a specific region of the image, and the feature algorithms are used to get quantitative information out of this region. There are three types of masks: default, combined, and function. Default masks contain all pixels that are detected as different from the background for each channel. Combined masks are created using Boolean logic to combine or subtract masks, and function masks are created with user input (e.g., spot, threshold, peak, and range) [38]. Once the mask defines the pixels of interest in the image, features are applied to calculate the quantitative information based on the signal intensity and location information. The eight feature categories are: size, location, shape, texture, signal strength, comparison, system, and combined (reference to the IDEAS manual). Singles cells are differentiated from doublets and debris by using the features aspect ratio (AR) (minor axis divided by the major axis of the default mask) and area (in microns squared corresponding to the magnification and number of pixels in the default mask) of the BF image. Single cells have an intermediate area value and a high AR. Doublets have larger areas and a lower AR than single cells. Debris has small areas and a range of ARs depending on the shape of the debris.

Another advantage of IFC is the high fluorescence sensitivity, which is a requirement for the detection of the smallest bacteria. This is achieved by reading out the signals of the CCD camera using time delay integration (TDI). In TDI, the photo charges stored in each image pixel are shifted from row to row down the detector, as each row of pixels is read sequentially off the bottom of the chip. Since the objects are continuously illuminated, the signals are integrated over the entire length of the detector with no readout noise, leading to a higher net sensitivity [39].

Flow cytometric methods, both conventional and imaging flow cytometry (IFC), still rarely appear as the methodology of choice for biofilm analyses. Thus, techniques that allow the evaluation and measurement of the physical and chemical aspects of biofilms as holistically treated structures prevail. Most of them do not measure the cellular characteristics of “key players” in the formation of biofilms or microorganisms [40]. The most widely used technique for assessing the viability of microbial cells from biofilms is based on colony counting to determine the colony-forming units (CFU) on an agar medium.

The approach presented in this study responded to the most serious limitations of the total viable count (poor plate) methods: (i) the fraction of live cells detached from the biofilm structure may not be representative of the original biofilm population, and (ii) the biofilm-derived cell subpopulation may belong to living but non-cultured forms of microorganisms, and as a result, it may not be detected by colony counting methods [41]. The developed tool, being a combination of imaging flow cytometry and a machine learning module-based interpretation protocol, significantly improved the analytic potential of conventional flow cytometry. The IFC instrument (a unique combination of a conventional flow cytometer with a fluorescence microscope), together with advanced software for an in-depth analysis of the cytometric results, provides the ability to image large numbers of cells in high resolution and improves the cytometric analysis of cells even in complex cell populations/consortia [42,43].

The key aspect is the software used to process, present, and interpret the results of the analysis by flow cytometry with bioimaging. They constitute correlated data of the measured parameters of fluorescence light with digitally processed images of the analyzed objects/cells. An image analysis is performed using software that allows the measurement of almost 1000 photometric and morphometric features [44]. Imaging flow cytometry increases the analytical potential and the ability to detect the functional and structural complexities within cell agglomerates, which implicates the improved precision of the determination of microbial contamination of food-processing lines and products [45]. The change in the “approach” to the analysis of cells using imaging cytometry is fundamental and results not only from the quantity or quality of the information obtained but also from the way/ways of interpreting and classifying the obtained data, hence the large role of interpretation tools, which include the machine learning module (ML) used at work to improve the resolution of the results. This resulted in an increase in the precision of determinations, organization of data, and determination of the most important trends in the interpretation of the results.

The ML-based software-assisted IFC analysis provided: (i) a precise “look” into the heterogeneity of the microbial population while increasing the range of information obtained about cells (direct correlation of the fluorescence intensity with cell morphology) [43] and (ii) allowed for comprehensive monitoring of the dynamics of complex processes, such as biofilm formation [43,46].

The application potential of the designed tool for the food industry seems to be related to crucial aspects of quality control and safety in the food production chain. The role of aggregates of microbial cells, ignored due to the limitations of detection of most instrumental analytical techniques, is an example of a challenge for such modern solutions, as presented in this study. Aggregates of microbial cells appear in the industrial environment in a “natural” way, as a stage in the biofilm life cycle, but also, their occurrence is associated with the cleaning and disinfection procedures routinely used in the food industry. They lead to the dispersion of biofilms and the release of aggregated forms of microorganisms on a scale exceeding the aggregate dispersion characteristics of the processes related to “natural” biofilm life cycles [19,23].

Recently, biofilm dispersal has been suggested as a promising strategy for increasing the contact surface for the action of biocides [19]. On the other hand, however, there is a concern about the increased emissions of aggregates, which are invasive forms that facilitate the spread of microorganisms along the production line [47]. In this way, undesirable and, especially, pathogenic microorganisms can be directly released into the production environment. In addition, bacterial aggregates are still characterized by an increased resistance to external factors, including cleaning and disinfection procedures, used in the food industry. It has been proven that microbial cells released from a biofilm may be more virulent, which is explained by decreased levels of the c-di-GMP-signaling molecule [48]. Such cells show a stronger ability to penetrate and kill macrophages [49].

## 4. Methods

### 4.1. Preparation of Samples for Analysis

Samples were collected from the surfaces of 3 diagnostic points: a mushroom crate (plastic surface), conveyor belt (stainless steel surface), and mushroom feeder (stainless-steel surface) from a fruit and vegetable processing company in Greater Poland. The analysis was carried out in 2 independent experiments (time points): samples A—5 August 2019 and samples B—20 September 2019. In that period, button mushrooms (*Agaricus bisporus*) were the processed product. As the company has a 3-shift working system, with cleaning procedures employed after each shift, representative samples were collected in the middle of each shift. In the collection procedure, 3 separate swabs of an area of 100 cm^2^ were taken using a sterile cotton swab. The swabs were immediately placed in tubes with 2 mL of 1% PBS solution and transferred to the Department of Biotechnology and Food Microbiology laboratory for direct staining and analysis.

### 4.2. Imaging Flow Cytometry Analysis to Assess the Structural and Functional Complexities of Biofilm-Derived Bacterial Aggregates 

The microbial cells in sample tubes were vortexed to detach the collected cells from inner and outer swab surfaces. Afterwards, a filtration step was employed using a nylon net 60-µm syringe filter (assembled with a Swinnex filter holder of 25 mm, both from Merck Millipore, Darmstadt, Germany). The samples were stained with the BacLight™ Redox Sensor™ Green Vitality Kit (Life Technologies, Carlsbad, CA, USA) according to the manufacturer’s instructions. The kit contains the membrane impermeable DNA dye propidium iodide (PI) (maximum excitation/emission at 493/636 nm) for cellular viability assessment and RedoxSensor Green (RSG) reagent to measure the metabolic activity. This compound is a fluorogenic redox indicator dye, which is subjected to conversion by microbial reductases involved in the electron transport chain [50]. The converted dye, following excitation (maximum = 490 nm), emits a green fluorescence (maximum = 520 nm). The intensity of the green fluorescence emission is directly proportional to the cellular redox potential, which reflects the levels of metabolic activity of microbial cells [51]. To assess the cellular metabolic activity and viability, the samples were acquired on an Amnis^®^ brand FlowSight^®^ (Luminex, Austin, TX, USA) imaging flow cytometer equipped with three excitation lasers (120 mW 405 nm, 60 mW 488 nm, and 150 mW 642 nm); a 70-mW 785-nm side scatter (SSC) laser; and a 12-channel CCD camera for signal detection. Only the 488-nm and SSC laser were enabled during acquisition (at 15–30 and 70 mW, respectively). The cells were run in PBS (pH 7.4, without calcium and magnesium ions) (Thermo Fisher Scientific, Waltham, Massachusetts, USA) as the sheath fluid and imaged at 20× magnification. Single-color compensation controls for PI and RSG were also acquired using the integrated software INSPIRE^®^ (Luminex, Austin, TX, USA) for data collection.

### 4.3. Machine Learning Based Interpretation Protocol of the Imaging Flow Cytometry Data

The image analysis was performed using image-based algorithms in ImageStream Data Exploration and Analysis Software (IDEAS^®^ 6.2.187, Luminex, Austin, TX, USA). Typical files contained imagery for 5000 events. The analysis was restricted to stained cells in the best focus. Stained cells were identified by their maximum pixel intensity (Raw Max Pixel > 100 (au)) for PI and RSG, which was confirmed by the corresponding images. Out-of-focus events were excluded by using the feature BF Gradient RMS, a measurement of image contrast. The analysis enabled the detection of nonactive (dead), intermediate (mid-active), and active microbial cells representing the discrete subpopulations. Subpopulations active, mid-active, and dead were defined based on differences in the level of metabolic activity measured as the cellular redox potential (CRP) by gating in the dot plots of green fluorescence from the RedoxSensor Green reagent (Ch02) vs. red fluorescence (channel 5: Ch05) from propidium iodide. The calculation of the CRP values was performed using medians of the green fluorescence (channel 2: Ch02) signals of the gated subpopulations [52]. 

To improve the differentiation of single cells (S), small aggregates (SA), and large aggregates (LA), and to make full use of the high dimensionality of the imaging data, we loaded the data into the machine learning (ML) module of IDEAS^®^ 6.3. After manual selection of the three “truth” populations, the ML algorithm calculated three super features (classifiers) that maximally separated each “truth” population from the others. The classifiers were based on user-defined and/or ML-generated features, which were normalized (each feature has a mean µ = 0 and a standard deviation σ = 1) and ranked by the Fisher’s discrimination ratio (RD) metric (RD = interclass variance/intraclass variance = [(µ(positive truth) − µ(negative truth)]/(σ (positive truth) + σ (negative truth))) to quantify the separation power. Afterwards, the classifiers were created by linearly combining the top-ranked features with weights proportional to the RD scores until a maximum separation was achieved (linear discriminant analysis).

For every classifier, truth populations between 35 and 50 events were manually tagged and loaded into the ML module. ML generated and tested the features of all 8 main categories corresponding to the BF, SSC, PI, and RSG channels. The resulting S_classifier contained a series of 10 differentially weighted features. They mainly described the elongation and size of the BF images and the homogeneity of the SSC signals. Singlets were smaller and less elongated, and the SSC signal was less homogenous than in the aggregates. The SA_ and LA_classifiers contained 119 and 45 features, respectively. The SA_classifier refined the shape of the aggregates in the BF image deeper by using more shape-related features, whereas the LA_classifier mainly used size features of the BF and SSC images. Thus, the small aggregates were more elongated and had a higher tendency to have a single axis of elongation than the singlets and large aggregates. The large aggregates themselves were defined by the highest diameters and areas of the SSC and BF images. All three classifiers were plotted into histograms and events with values higher than zero belonging to the images that were best-represented by their classifier.

### 4.4. Statistical Analysis

Obtained results were expressed as the mean ± standard deviation (±SD) of the replicates. The statistical importance of the differences between compared sets of data was analyzed using one-way analysis of variance (ANOVA). The levels of significance were set at *p* < 0.05. Statistical analysis and graphical presentation of the obtained data were carried out using OriginPro, Version 2021b (OriginLab Corporation, Northampton, MA, USA).

## 5. Conclusions

Imaging flow cytometry increases the analytical potential and the ability to detect functional and structural complexities within cell agglomerates, which increases the precision of the determinations of microbial contamination of technological lines and products. Optimizing the analytical procedures in a combination of fluorescence staining and imaging flow cytometry has the potential to improve the detection of important diagnostic points within the food production chain (food-processing hot spots). They may constitute areas prone to biofilm formation and/or surfaces where biofilm dispersal initiates, resulting in the spread of numerous bacterial aggregates. The assessment of the intensity of the cell agglomerates in combination with detailed morphological (small- and large-cell aggregates) and functional characteristics (detection of active, indirect metabolic activity and inactive cells) is a perspective for the development of microbiological diagnostics.

The developed tool provided more detailed characteristics of bacterial aggregates within biofilm structures combined with high-throughput screening potential. The IFC procedure for determining the cellular redox potential of the microorganisms forming biofilms on food-processing surfaces enabled a “look” inside the three-dimensional structure of individual aggregates and determined the physiological state of each of the cells that make up the aggregate and single forms.

## Figures and Tables

**Figure 1 molecules-26-07096-f001:**
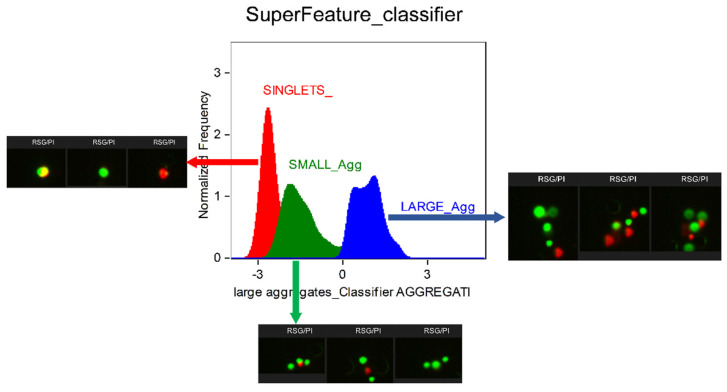
Characteristics of microbial aggregates derived from biofilms occurring on industrial surfaces and analyzed with the use of imaging flow cytometry (IFC) assisted by an advanced tool for processing and interpreting the cytometric results: a machine learning (ML) module of IDEAS software. The combination of IFC with the ML-based interpretation of flow cytometric data allowed for the precise discrimination of single microbial cells (singlets) from cellular complexes (aggregates), further characterized to identify small (composed of 2 to 3 microbial cells) and large aggregates (more than 3 cells).

**Figure 2 molecules-26-07096-f002:**
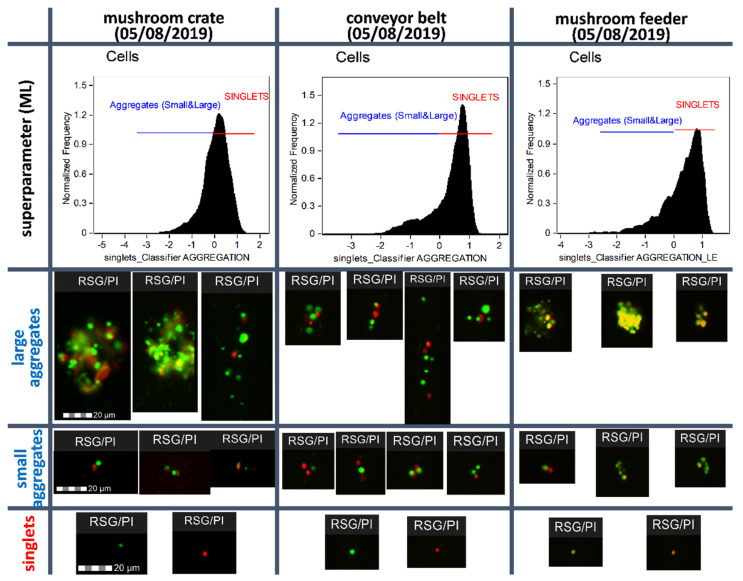
Samples from time point 5 August 2019 of food-processing surfaces analyzed using the imaging flow cytometry protocol assisted by a machine learning (ML) module. The analysis demonstrated significant differences in the structural and functional complexities of the microbial aggregates associated with biofilms. Histograms show the intensity values of the classifier (super feature) generated by the ML module to discriminate singlets vs. small and large microbial aggregates. Higher percentages of microbial aggregates were associated with the higher average metabolic activity of bacterial cells forming biofilm structures. This indicated the protective role of cellular aggregates—the degree of aggregation affected the survival of biofilm-associated microbial cells.

**Figure 3 molecules-26-07096-f003:**
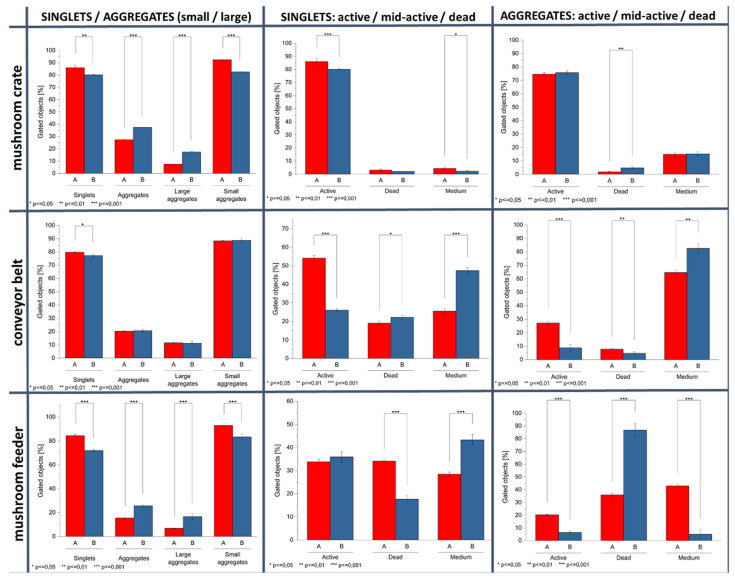
Distribution of the biofilm-derived microbial subpopulations defined using the imaging flow cytometric analysis combined with a machine learning (ML) module for the interpretation of the cytometric results. The samples were collected from the surfaces of 3 diagnostic points of food-processing surfaces. The analysis was carried out in 2 independent experiments (time points): samples A—5 August 2019 and samples B—20 September 2019. The microbial aggregates and singlets from the biofilm samples were simultaneously examined for the percentages of active, mid-active, and nonactive (dead) cells using the measurements of the metabolic activity of the microbial cells (evaluation of the cellular physiology). Asterisks over brackets indicate a significant difference between samples (* *p* < 0.05, ** *p* < 0.01, and *** *p* < 0.001). Whiskers are standard deviations (SDs).

## Data Availability

Data available on request due to restrictions eg privacy or ethical.

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
