# Peer review of "Imaging Flow Cytometry to Study Biofilm-Associated Microbial Aggregates"

_molecules, 2021, doi:10.3390/molecules26237096_

Round 1

Reviewer 1 Report

The study highlights the analytical tool that is designed for the characterization of microbial aggregates from biofilms on food processing surfaces. Using the flow cytometry to characterize and image the aggregates is interesting, however, there is lack of discussions on the results sections. The discussions sections should be improved by highlighting the rationale behind the key findings of the paper.

Below are the general and specific comments:

Line 18, 157-missing point in 75,3%

Line 52-62. Not clear sentences. The sentences can be broken up into two and rewritten to make it easy to understand.

Line 63-65. Not clear, need to paraphrase. Check grammar as well. “Not” would be after “limited”.

Line 67. “Biofilm dispersing into bacterial aggregates” might be better.

Line 68-76. Unclear sentences. Check the grammar. Extra space in line 72

Line 79-82. Grammar is not correct. Sentence is too long to understand the message.

Line 84-94. Very difficult to understand. Need to rewrite again with correct grammars.

Introduction. It is not quite clear what author’s research questions are. Is it solving problems that occurs from biofilms or from aggregates (dispersed from biofilms as per the authors)? The overall research questions should be stated clearly.

4.2. Extra space in line 338 and 342. Comma should be after viability in line 340. Heading should be shorter if possible.

line 380, 240, 244. Extra space

Figure 1. The caption for fig 1 is rewritten from results section (2nd paragraph). Instead of repeating, making caption short would read more clearly.

Line 151-153. Unclear sentence. Instead of writing in brackets, making it following sentence from previous would be appropriate.

Figure 2 and 3. Caption should be shorter. Majority of the caption is already stated in M&M so restating is not required.

Results. In the sample, some of the cells might be live but not metabolically active so might not be stained by the assay. How does authors explain those populations?

Line 173. Missing points in 37,4%

Discussion. The authors stated about aggregates dispersing from biofilm. Does the author have any evidence to show that the swabbed sample are aggregates dispersed from biofilm like staining for different biofilm components?

The cells in the fluorescent images (fig 2) does not seem to attach together in most of the images. Is there any rationale for why cells are not attaching together in aggregates.

Scale bar should be given in the fluorescent images.

I appreciate the explanation of flow cytometry and ML in the discussion section. However, results should also be discussed.

Line 169-176. Can authors explain the rationale behind difference in aggregates percentage between the mushroom crate and conveyor belt and mushroom feeder? Is it materials specific?

Author Response

Point 1: Line 18, 157-missing point in 75,3%

Response 1: corrected

Point 2: Line 52-62. Not clear sentences. The sentences can be broken up into two and rewritten to make it easy to understand.

Response 2: the mentioned fragment has been redrafted.

Point 3: Line 63-65. Not clear, need to paraphrase. Check grammar as well. “Not” would be after “limited”.

Response 3: the mentioned sentence has been redrafted.

Point 4: Line 67. “Biofilm dispersing into bacterial aggregates” might be better.

Response 4: corrected

Point 5: Line 68-76. Unclear sentences. Check the grammar. Extra space in line 72

Response 5: corrected

Point 6:Line 79-82. Grammar is not correct. Sentence is too long to understand the message.

Response 6: the mentioned sentence has been redrafted

Point 7: Line 84-94. Very difficult to understand. Need to rewrite again with correct grammars.

Response 7: the mentioned fragment has been redrafted.

Point 8: Introduction. It is not quite clear what author’s research questions are. Is it solving problems that occurs from biofilms or from aggregates (dispersed from biofilms as per the authors)? The overall research questions should be stated clearly.

Response 8: In our opinion the key research question behind the undertaken subject has been stated clearly as the 5th paragraph of Introduction section: 

"The presence of biofilms on various technological surfaces that come into contact with food (from the raw material to the final product) is a constant source of microbial contamination (Abdallah et al., 2014). Thus the problem of biofilms dispersal in industrial conditions (especially in the food industry) requires separate attention, especially in view of the increasing share of industrial processes in world food production (Galié et al., 2018)."

Of course if needed we may still improve it.

Point 9: 4.2. Extra space in line 338 and 342. Comma should be after viability in line 340. Heading should be shorter if possible.

Response 9: Corrected and the heading significantly redrafted.

Point 10: line 380, 240, 244. Extra space

Response 10: corrected

Point 11: Figure 1. The caption for fig 1 is rewritten from results section (2nd paragraph). Instead of repeating, making caption short would read more clearly.

Response 11: The caption of the mentioned Figure has been significantly redrafted (repetitions removed).

Point 12: Line 151-153. Unclear sentence. Instead of writing in brackets, making it following sentence from previous would be appropriate.

Response 12: The whole senetnce has been redrafted (divided into 2 to improve readability and understanding).

Point 13: Figure 2 and 3. Caption should be shorter. Majority of the caption is already stated in M&M so restating is not required.

Response 13: The captions of both mentioned Figures has been significantly redrafted (duplicates from Materials and Methods section has been removed).

Point 14: Results. In the sample, some of the cells might be live but not metabolically active so might not be stained by the assay. How does authors explain those populations?

Response 14: The mentioned cellular fraction may represent the sub-population of mid-active microbial cells demonstrating the intermediate level of cellular metabolic activity. The applied measurement activity of cellular enzymes from oxido-reductases family (redox sensor reagent) enabled the redefinition of the term cellular viability (it is based on cellular metabolism). This provided an alternative to the commonly used parameter of distinguishing between live and dead cells - cellular integrity, which gives information about cellular viability. Thus, our approach enabled to investigate the vitality of microbial cells. This contrasts with the evaluation of membrane intergrity (live/dead discrimination) or the ability of microbial cells synthesize macromolecules (a prerequisite of cell division - evaluation of the ability of cells to growth on culture media). In our previous study the occurance of microbial forms in mid-active physiological state was investigated together with their growing potential (https://doi.org/10.1080/08927014.2018.1508569).

Point 15: Line 173. Missing points in 37,4%

Response 15: corrected.

Point 16: Discussion. The authors stated about aggregates dispersing from biofilm. Does the author have any evidence to show that the swabbed sample are aggregates dispersed from biofilm like staining for different biofilm components?

Response 16: In our previous work the similar working surfaces were analysed using the conventional flow cytometry and focusing only on singlets (single bacterial cells). We used different protocol to disintegrate biofilm fragments and cellular aggregates - swabs after collection were sonicated in 1% PBS solution. This led to the disruption of aggregates to release microbial cells. In the revised study we needed to alter the protocol in order to leave the aggregates untouched.

Furthermore, the nature of flow cytometric analysis (in particular the discussed below mechanism of hydrodynamic focusing imployed within cytometric flow chamber) causes the cells in analysed suspension (sample) to travel through cytometric observation as separate events. Only aggregated cells (assembled together) or clumps of cellular (and non-cellular) debris are able to flow relatively intact.

Point 17: The cells in the fluorescent images (fig 2) does not seem to attach together in most of the images. Is there any rationale for why cells are not attaching together in aggregates.

Response 17: Our analysis performed using imaging flow cytometer demonstrated the singlets of bacterial cells and variety of bacterial aggregated forms. The basics of flow cytometry mode of action is the hydrodynamic focusing which drive the cells from sample to travel one after another. This is the sheath fluid pressure driven movement of cells within a flow chamber. The sheath fluid pressure induces cells to disperse into singlets rather then clip them together, so when aggregates are observed this means the cells forming an aggregate are well "assembled". In the case of biofilms this means the extracellular matrix (e.g. exopolymeric substances - EPS), which allow bacteria to stick to each other. The EPS is "the glue" which is not exclusively observed on images gallery generated by the instrument.

Point 18: Scale bar should be given in the fluorescent images.

Response 18: Corrected - the Figure 2 has been suplemented with cell images with incorporated scale bars.

Point 19: I appreciate the explanation of flow cytometry and ML in the discussion section. However, results should also be discussed.

Response 19: The Results section has been supplemented with the appropriate fragment discussing key aspects of the obtained results.

Point 20: Line 169-176. Can authors explain the rationale behind difference in aggregates percentage between the mushroom crate and conveyor belt and mushroom feeder? Is it materials specific?

Response 20: This can be explained, to some extent, as material specific but the the main reason behind this difference is the position of the specific element on production line (earlier or later in the sequence of processing event). The samples originated from a fruit and vegetable processing company in Greater Poland. The tested elements of the technological line (mushroom crate, conveyor belt, mushroom feeder) appear in different places of the line. The mushroom crate is positioned at the begining of the processin line while conveyor belt and mushroom feeder occur later in processing sequnce). This may result in higher microbial contamination rate accompanied by higher percentages of active bacterial forms. These lines were previously examined for microbial contamination using conventional flow cytometric approach combined with specific sorting of bacterial cells from the defined sub-populations (see: http://dx.doi.org/10.1080/08927014.2016.1201657).

Reviewer 2 Report

Dear authors,

I think your work is important and well done. My questions and comments would be the following:

Line 43: Many biofilms are surface-associated, but not all. Thus, I would deleate "surface-associated

Line 46-48: Did you look for more recent publications? Because biofim-related research has increased during the last years tremendously there might be more recent studies published which should be included.

Line 59-63: Sentence is long and very difficult to read. Maybe you can try to find a new way to write this a bit easier to understand.

Line 67: Not only aggregates are dispersed, but planktonic cells as well.

Line 70: Not only the densitiy of the population but environmental conditions (e.g. temperature, pH, oxygen and several other factors) and age of the biofilms (e.g. accumulation of metabolites) influence dispersion of biofilms or release of single bacteria.

Line 85 and 91 are very similar

The results-section is not very comprehensible without the methodes. Maybe Methods could be placed after the introduction?

Line 150 and 151: To my knowledge, bacteria within the biofilm do not show higher but usually a reduced metabolic activity (aggregation and biofilm formation lead to homeostasis within the biofilm, meaning the bacteria need less energy and less activity). This increased activitiy of bacteria in biofilms is mentioned several times in the manuscript. Or do I get this wrong and you mean, that the activity of the bacteria-aggregates cumutales, leading to a higher level of activity in total compared to single cells..?

Figure 2: Please, clarify the labels in the top row graphs.

Figure 3: It would be better to have the identical order in the heading and in the labels below and maybe the identical terms as well (e.g. "mid-active" vs "Medium"); the text within the graphs (especially the singificances given) is really very small and hardly readable in a printout; "Whiskers" are mentioned for standard deviations - I can't find any standard deviations..?

Throughout the manuscript are some blanks too much or missing - this should be improved.

Author Response

Point 1: Line 43: Many biofilms are surface-associated, but not all. Thus, I would deleate "surface-associated

Response 1: Corrected - indicated fragment has been deleted.

Point 2: Line 46-48: Did you look for more recent publications? Because biofim-related research has increased during the last years tremendously there might be more recent studies published which should be included.

Response 2: Corrected - Introduction and Results section have been suplemented with some of more recents publications.

Point 3: Line 59-63: Sentence is long and very difficult to read. Maybe you can try to find a new way to write this a bit easier to understand.

Response 3: Corrected

Point 4: Line 67: Not only aggregates are dispersed, but planktonic cells as well.

Response 4: Agree, but the work deal with biofilm forming microbial cells, that is why such statement seem authorised.

Point 5: Line 70: Not only the densitiy of the population but environmental conditions (e.g. temperature, pH, oxygen and several other factors) and age of the biofilms (e.g. accumulation of metabolites) influence dispersion of biofilms or release of single bacteria.

Response 5: Agree, but still a focus on cellular density does not rule out the participation of other factors.

Point 6: Line 85 and 91 are very similar

Response 6: Agree - the second sentence (in fact a repetition) has been deleted.

Point 7: The results-section is not very comprehensible without the methodes. Maybe Methods could be placed after the introduction?

Response 7: I agree but the Materials and Methods sections has been placed at the end of the manuscript according to Journal requirements. If Editor agrees we may gladly shift this section and place it in front of the Results section.

Point 8: Line 150 and 151: To my knowledge, bacteria within the biofilm do not show higher but usually a reduced metabolic activity (aggregation and biofilm formation lead to homeostasis within the biofilm, meaning the bacteria need less energy and less activity). This increased activitiy of bacteria in biofilms is mentioned several times in the manuscript. Or do I get this wrong and you mean, that the activity of the bacteria-aggregates cumutales, leading to a higher level of activity in total compared to single cells..?

Response 8: Agree, there are however a few explanations for the occurrence of sub-population of active microbes within biofilm derived cells inside aggregates:

-) the self-induced cell lysis during biofilm development can play an important role in biofilm dispersal and ecology. The autocidal activity may also result in the use of dead cells as a nutrient source for surviving bacteria.

-) transition from active metabolism to maintenance metabolism, which at the expense of growth cessation retain a relatively high level of metabolic activity to support core metabolism.

The problem has been widely discussed in our previous work: https://doi.org/10.1080/08927014.2018.1508569

Point 9: Figure 2: Please, clarify the labels in the top row graphs.

Response 9: The labels of histograms (graphs) has been explained in the caption text:

"Histograms show the classifier (superfeature) generated by ML module to discriminate singlets vs small and large microbial aggregates."

Point 10: Figure 3: It would be better to have the identical order in the heading and in the labels below and maybe the identical terms as well (e.g. "mid-active" vs "Medium"); the text within the graphs (especially the singificances given) is really very small and hardly readable in a printout; "Whiskers" are mentioned for standard deviations - I can't find any standard deviations..?

Response 10: The Figure 3 would have to be divided into 2 or 3 separate figures to gain more readability - this will require a huge reorganisation of data. It can be done but requires an extra time and has to be well studied.

As for the SD and Whiskers showing them - they really do exist on the Figure. We may always provide the raw statistic data.

Point 11: Throughout the manuscript are some blanks too much or missing - this should be improved.

Response 11: The whole manuscript has been reviewed for the occurance of numerous blanks - issues fixed.

Reviewer 3 Report

Greetings,

the paper as it is does not appear to require revision, in my opinion.

the MS is very well written I think it is ready for publication.

I can find only one specific comment: the references in the MS are not numbered. 

Kind regards

Author Response

Point 1: I can find only one specific comment: the references in the MS are not numbered.

Response 1: Please be more specific about this issue - is it the references in the text that should be numbered? Of course we are always ready to change that.

Reviewer 4 Report

        The designed analytical tool of this paper is novel and significant in assessing the complexity of microbial aggregates from biofilms. However, there are some problems to be further improved as well. My detailed comments are as follows:

1. The results of the study are not clear to the reader. For example, why the authors did 2 independent experiments (time points): samples A – 05/08/2019 and samples B – 20/09/2019 ? An explanation should be provided in more detail.

2. In addition, the legend in the figure 3 is out of order.

Author Response

Point 1: The results of the study are not clear to the reader. For example, why the authors did 2 independent experiments (time points): samples A – 05/08/2019 and samples B – 20/09/2019 ? An explanation should be provided in more detail.

Response 1: The developed method (protocol) revealed high level of sensitivity as it detected subtle differences in how the line was handled by different employees, as was the case here. Considering differences between 2 presented time points (samples A – 05/08/2019 and samples B – 20/09/2019) tested food-processing line differed in terms of the moment of carrying out the cleaning and disinfection procedures and in terms of the human factor (teams performing procedures).

Point 2: In addition, the legend in the figure 3 is out of order.

Response 2: The figure captions has been significnatly redrafted.

Reviewer 5 Report

The manuscript is very well written, structured and represent an important contribution to biofilm science. Howsoever the authors should explain the following issue:

Figure 3: Authors are statistically comparing two independent experiments in time points “A” and “B” and found statistically significant differences between experiments. That means and experiments and methods that they are proposing are not repeatable? The authors should explain the aim of statistical analysis and results.

Author Response

Point 1: Figure 3: Authors are statistically comparing two independent experiments in time points “A” and “B” and found statistically significant differences between experiments. That means and experiments and methods that they are proposing are not repeatable? The authors should explain the aim of statistical analysis and results.

Response 1: It has been demonstrated that the developed method (protocol) is highly sensitive as it detects even subtle alterations in measured cellular parameters of microbes dwelling on tested surfaces. This accuracy refers to identify differences in the handling of food-processing lines by different employees. This was the case here - tested food-processing line differed in terms of the moment of carrying out the cleaning and disinfection procedures and in terms of the human factor in the form of teams performing procedures. This may affect the contamination levels and explain differences between 2 presented time points (samples A – 05/08/2019 and samples B – 20/09/2019).